# The Single Intra-Articular Injection of Platelet-Rich Plasma vs. Non-Steroidal Anti-Inflammatory Drugs as Treatment Options for Canine Cruciate Ligament Rupture and Patellar Luxation

**DOI:** 10.3390/vetsci10090555

**Published:** 2023-09-04

**Authors:** Kristina Raulinaitė, Rasa Želvytė, Kristina Škėmienė, Evelina Burbaitė, Birutė Karvelienė, Ingrida Monkevičienė

**Affiliations:** 1Department of Anatomy and Physiology, Faculty of Veterinary, Veterinary Academy, Lithuanian University of Health Sciences, Tilžės Str. 18, 47181 Kaunas, Lithuania; rasa.zelvyte@lsmu.lt (R.Ž.); ingrida.monkeviciene@lsmu.lt (I.M.); 2Laboratory of Biochemistry, Neuroscience Institute, Lithuanian University of Health Sciences, Eivenių Str. 4, 50161 Kaunas, Lithuania; kristina.skemiene@lsmu.lt; 3San Marco Veterinary Clinic, Neurology and Neurosurgery Division, Viale dell’Industria 3, 35030 Padova, Italy; 4Dr. L. Kriaučeliūnas Small Animals Clinic, Faculty of Veterinary, Veterinary Academy, Lithuanian University of Health Sciences, Tilžės Str. 18, 47181 Kaunas, Lithuania; birute.karveliene@lsmu.lt

**Keywords:** dog, cranial cruciate ligament rupture, patellar luxation, platelets rich plasma, nonsteroidal anti-inflammatory drugs, tumor necrosis factor alpha

## Abstract

**Simple Summary:**

Cranial cruciate ligament rupture and patellar luxation are among the most common orthopedic pathologies in canine patients. Currently, non-steroidal anti-inflammatory drugs are widely used for managing those cases. Platelet-rich plasma is one of the newest non-invasive treatment options and possibly is more beneficial than conventional treatment. This study’s general aim was to determine the difference between the effects of single intra-articular injection of platelet-rich plasma and oral non-steroidal anti-inflammatory drugs usage on inflammatory cytokine tumor necrosis factor alpha levels and clinical outcome in cases of canine cranial cruciate ligament rupture and patellar luxation. To achieve this, we treated named pathologies differently (using non-steroidal anti-inflammatory drugs or platelet-rich plasma) and measured clinical outcomes and concentration of tumor necrosis factor alpha in the patient’s blood serum during a period of 28 days. The results of our study suggest that platelet-rich plasma can be used in treating canine cruciate ligament rupture and canine patellar luxation effectively and is more beneficial than the usage of non-steroidal anti-inflammatory drugs.

**Abstract:**

Cranial cruciate ligament rupture (CCLR) and patellar luxation (PL) are common pathologies affecting canines. Non-steroidal anti-inflammatory drugs (NSAIDs) are commonly used as a non-surgical treatment plan in these cases. Clinical usage of platelet-rich plasma (PRP) is an emerging area of interest in veterinary medicine. There is a lack of studies comparing those two different treatment methods in veterinary medicine. The main purpose of this study was to evaluate and compare the use of oral NSAIDs and single intra-articular injection of PRP on treatment outcomes in cases of canine CCLR and PL. Dogs diagnosed with CCRL (*n* = 12) and PL (*n* = 10) were subgrouped by the severity of pathologies and administered treatment: half of the CCRL and PL groups were orally administered NSAIDs and supplements for 14 days, and the other half received a single intra-articular PRP injection into affected stifle joint. We measured serum TNF-α levels and clinical outcomes (lameness scores, painfulness to manipulations, goniometry of stifle joint in flexion and extension, and muscle strength) before treatment, at day 14 and day 28 of treatment. The results of TNF-α concentration indicates a significant difference between groups of differently treated partial CCLR groups on d14 (*p* = 0.006). Results of group CCLR-P1 on d14 were decreased, while results of group CCLR-P2 on d14 were increased. When comparing TNF-α concentration between all CCLR cases treated with NSAIDs and treated with PRP, there was a significant difference between those groups on d14 (*p* = 0.001). The results of TNF-α concentration indicates a significant difference between groups of differently treated PL-III on d28 (*p* = 0.036). Results of group PL-III1 indicate growth of TNF-α concentration, while at the same d28, results of group PL-III2 indicate decreased levels of cytokine, comparing results between the subgroups at the same time point and within subgroups from baseline. Results indicate a significant difference in muscle strength between group CCLR-P1 and group CCLR-P2 on d28 (*p* = 0.007), indicating an increment in muscle strength in group CCLR-P1 up to d14 and its reduction up to d28, and muscle strength of group CCLR-P2 increasing up to d28. When comparing the muscle strength between all CCLR cases treated with NSAIDs and treated with PRP, there was a significant difference between those groups on d28 (*p* = 0.007). In conclusion, a single intra-articular injection of PRP has a superior effect on management of inflammatory processes, has better clinical outcomes, and longer duration of action than oral NSAIDs, in the treatment of canine CCRL or PL.

## 1. Introduction

Cranial cruciate ligament rupture (CCLR) is the most common pathology leading to pelvic limb lameness and stifle osteoarthritis (OA) in dogs [1,2,3,4]. CCLR can be partial or complete [5,6]. Some research recommends non-surgical treatment of CCLR for dogs weighing <15 kg [7,8]. Nevertheless, numerous studies have shown that the surgical treatment is better and more recommended than non-surgical treatment option [5,9]. Non-steroidal anti-inflammatory drugs (NSAIDs) are mostly used in combination with glucosamine/chondroitin sulfate supplementation for non-surgical treatment of CCLR [10].

Patellar luxation (PL) is one of the most common orthopedic diseases that can be con-genital or traumatic in its origin. Congenital PL is resulting from the quadriceps mechanism malalignment and inadequate trochlear groove that leads to the development of degenerative joint disease and painfulness in canine patients [11,12,13,14]. PL can be medial, lateral, or bidirectional, and medial being the most common form of luxations [15,16,17,18,19,20]. PL in canine is graded into four grades. Grade I—patella does not spontaneously luxate, but can be manually luxated, but afterwards returns into trochlear groove. Grade II—the patella may be manually displaced or may luxate with flexion/extension of the stifle; the patella remains luxated until manually reduced. Grade III—the patella remains luxated majority of the times but may be manually reduced; flexion/extension of stifle results in relaxation. Grade IV—the patella is permanently luxated and cannot be manually reduced [15]. PL can be treated surgically or medically. Non-surgical management is recommended in cases of animals that are not significantly affected clinically. This treatment plan usually includes the administration of NSAIDs in association, or not, with weight control and physical rehabilitation [17,21]. 

Tumor necrosis factor alpha (TNF-α) has an important role in joint inflammation and cartilage degradation in osteoarthritic joints [22]. With other cytokines, this pro-inflammatory cytokine initiates the inflammatory cascade and palliates disease progression [23]. TNF-α participates in articular cartilage degradation in cases of OA [24]. OA causes the expansion of mRNA and protein levels of inflammatory cytokines, including TNF-α. Pro-inflammatory cytokines can worsen the course of the disease, while anti-inflammatory cytokines can promote healing [23]. 

NSAIDs act in an anti-inflammatory manner because of the inhibitory effect on cyclooxygenase (COX) [25,26]. Cimicoxib (CX) is a highly selective COX-2 inhibitor that acts in an anti-inflammatory and analgesic manner [27]. Long-term NSAIDs therapy has a lot of unwanted adverse reactions such as gastrointestinal tract disturbance as well as renal, hepatic, and coagulation pathologies [28,29,30,31]. 

Platelet-rich plasma (PRP) is an autogenous fluid concentrate composed mostly of platelets and various growth factors. Recent studies and research have shown PRP to inhibit healing by providing growth factors, cytokines, chemokines, and other compounds [32,33,34,35]. Cytokines and growth factors effectively palliate inflammation and initiate the anabolic process and tissue regeneration [36]. The intra-articular administration of the PRP is thought to slow the progression of OA [37]. 

Due to lack of studies evaluating the impact and efficacy of intra-articular PRP injections as treatment option for orthopedic pathologies in dogs and comparing them with classic NSAID protocol, this study was conducted with the primary objective of comparing, evaluating, and determining differences and effects between the efficacy of oral NSAID administration and single intra-articular PRP injection on changes in TNF-α levels and clinical outcome in canine CCLR (partial and complete) and PL (grade II and III). 

Our hypothesis was that a single intra-articular PRP injection affects TNF-α levels and clinical outcomes in canine CCLR (partial and complete) and PL (grade II and III) as much or more than oral NSAID administration and is an effective treatment option for the given pathologies.

## 2. Materials and Methods

This study was conducted in the Dr. L. Kriaučeliūnas small animal clinic and integrated studies, science and business valley “Santaka” of Lithuanian University of Health Sciences (LSMU) between September 2019 and December 2021. The procedures complied with the criteria provided by the Lithuanian animal welfare regulations (No. B1-866, 2012; No. XI-2271, 2012) and the decree of the director of the State Food and Veterinary Service, the Republic of Lithuania (No. B6-(1.9)-2103, 2020). Owners were informed and consented for patient participation. The authors ensured that the data of the study participants were processed and protected in accordance with the laws of the Republic of Lithuania. The study was carried out in compliance with the EU legislation. 

Thirty-two dogs were included in the study. All patients (*n* = 22) enrolled in the study were mixed-breed companion dogs. Dogs were studied based on the inclusion criteria: no administration of NSAIDs and glucosamine/chondroitin sulfate supplements for two months before admission and physical, orthopedic, and diagnostic imaging evaluation confirming CCLR (partial or complete) or medial PL (grade II or III) and no concurrent diseases. Only patients with PL grades II and III were included in the study from all PL cases, as owners of dogs with PL grade I refused treatment, and all owners of patients with PL grade IV required recommended surgical treatment. All owners of enrolled dogs refused surgical treatment due to financial considerations and voluntarily chose non-surgical treatment option. Patients with concomitant stifle joint problems or orthopedic or neurologic conditions affecting the hind limbs were excluded from the study.

### 2.1. Study Groups

***CCLR group.*** The animals under study were twelve mixed-breed dogs (seven females, five males, mean weight 18.2 ± 7.5 kg, mean age 3.3 ± 1.1 years) diagnosed with CCLR due to the clinical (lameness, painfulness of manipulations, positive stifle drawer, and tibial compression tests) and radiological (signs of joint effusion, OA changes, and changes in tibial plateau angle) examinations findings. Partial and complete CCLR were determined by difference in orthopedic examination: if animal showed positive drawer test but negative tibial compression test, it was determined as having partial tear, while if animal showed positive tibial compression test, it was determined as having complete tear. This method of evaluation of CCLR was based only on orthopedic examination findings and is not sensitive enough as other diagnostic methods (ultrasound, MRI, etc.). The dogs were divided into two groups: a group of partial CCLR (CCLR-P; *n* = 6) and group complete CCLR (CCLR-C; *n* = 6). These two main groups were further divided into two subgroups based on the owners’ consent to a specific treatment plan suggested by the veterinarian: NSAIDs and supplements (group CCLR-P1 (*n* = 3) and group CCLR-C1 (*n* = 3)) or single intra-articular PRP injection (group CCLR-P2 (*n* = 3) and group CCLR-C2 (*n* = 3)). Clinical and orthopedic examinations and blood analysis for TNF-α level were performed in each group on the day before starting treatment (d0), fourteen days after starting treatment (d14), and twenty-eight days after starting treatment (d28).

***PL group.*** The animals under study were ten mixed-breed dogs (four females, six males, mean weight 5.5 ± 1.7 kg, mean age 2.5 ± 1 years) diagnosed with PL due to the clinical and radiological examination. Based on the grade of PL, dogs were assigned to two groups of five each: grade II (group PL-II (*n* = 5)) and grade III (group PL-III (*n* = 5)). These two main groups were further divided into two subgroups depending on the treatment plan: NSAIDs and supplements (group PL-II1 (*n* = 2) and group PL-III1 (*n* = 3)) and single intra-articular PRP injection (group PL-II2 (*n* = 2) and group PL-III2 (*n* = 3)). Clinical and orthopedic examinations and blood analysis for TNF-α concentration were performed in each group on the day before starting treatment (d0), fourteen days after starting treatment (d14), and twenty-eight days after starting treatment (d28).

***Treatment plan of CCLR and PL groups.*** Each dog of group CCLR-P1, CCLR-C1, PL-II1, and PL-III1 received CX (2 mg/kg, orally every twenty-four hours for fourteen days; Vétoquinol SA, France) and supplements of glucosamine/chondroitin (7.2 µg cyanocobalamin, 53.1 µg folic acid, 2,6 mg thamin, 0.51 mg riboflavin, 1.68 mg pyridoxine, 2.26 mg niacin, 0.71 mg vitamin E, 55.5 mg calcium, 44.6 mg phosphorus P, 52.5 mg omega 3 nutrients, 225 mg glucosamine sulphate 2KCl, 162 mg chondroitin sulphate, and 59.94 mg MSM per 22–33 kg due to manufacturer’s recommendation, orally every twenty-four hours for fourteen days; Bob Martin, UK). Each group of dogs that were treated with PRP injections received a single intra-articular PRP injection.

### 2.2. Performed Examinations and Tests

***Clinical and orthopedic examinations.*** Dogs were assessed at d0, d14, and d28 with clinical and orthopedic examinations. Mucous membrane color, capillary refill time, heart rate and sounds, breathing rate and sounds, rectal temperature, and superficial lymph nodes palpation were recorded. Degree of lameness (0–5), presence of stifle effusion, presence of cranial drawer and cranial tibial thrust, the painfulness of stifle manipulations, and patellar luxation were among the findings recorded. Stifle range was evaluated during goniometric measurements in flexion and joint extension [38]. Canine Muscle Strength Tests were performed for muscle strength evaluation. Each test was performed two times on standing animal: first time in normal standing position, the second time—while animal was standing with thoracic limbs on the incline. During the testing, one pelvic leg was lifted from the ground while evaluating the ability of a limb to maintain a static standing position. The muscle strength was graded into 0–5 grades [39]. Additional abnormal orthopedic findings were also recorded. All testing was performed and evaluated by the same veterinarian who was not blinded to the treatment groups. 

***Blood samples collection.*** Three samples per patient were collected three times. Blood was collected by performing a peripheral blood draw from the cephalic vein. Whole blood (4 mL) was collected into a blood sample collection tube with lithium heparin (BD Vacutainer, USA) and centrifuged at 1000 rpm for 15 min. After centrifugation, plasma was collected into 1.5 mL cryogenic tubes (Eppendorf, Hamburg, Germany) and stored at −80 °C immediately. Each blood draw was performed at d0, d14, and d28.

***Preparation and application of platelet-rich plasma*.** Each dog of groups CCLR-P2, CCLR-C2, PL-II2, and PL-III2 was sedated with 5 mcg/kg of dexmedetomidine hydrochloride and 0.4 mg/kg of butorphanol given intramuscularly. Autologous blood was obtained by performing a peripheral blood draw. Blood was drawn by clipping an area over the jugular vein, and then the skin was aseptically cleaned. Whole blood (15 mL) was extracted aseptically from a jugular vein and collected into a 15-mL ACP double syringe (Arthrex, Inc., Naples, FL, USA) according to the manufacturer’s instructions. The double syringes were centrifuged at 1500 rpm (350 g, ‘soft spin’) for five minutes. After aseptic preparation, the ACP (2 mL) was injected into the damaged stifle joint using 1.5-inch 20-gauge needle. 

***ELISA.*** The concentrations of TNF-α in dog serum samples were measured using an antibody-based, sandwich enzyme-linked, immunosorbent assay (ELISA) kit (Abbexa^®^, Houston, TX, USA) strictly according to the procedures recommended by the manufacturer. Briefly, the assay was a two-site sandwich ELISA that offers optimal sensitivity as low as <6.1 pg/mL, with a range of detection of 15.6–1000 pg/mL. The 96-well microplate provided in the kit, which had been pre-coated with an antibody specific for TNF-α, was used to capture the soluble TNF-α from all samples (‘capturing’ antibody). Diluted standards (100 μL) were placed onto pre-coated wells as diluted samples (100 μL) and incubated for 1 h at 37 °C. Then, liquids were discarded, 100 μL of Detection Reagent A was aliquot into each well, and all samples were incubated for 1 h at 37 °C. After the incubation, each well was washed with wash buffer three times. After washing procedure, 100 μL of Detection Reagent B was added to each well, and all samples were incubated for 30 min at 37 °C. The wash process was repeated five times. A 90 μL volume of TMB Substrate was added into each well, mixed thoroughly, and incubated at 37 °C for 15 min. Any exposure to light was avoided. A 50 μL volume of Stop Solution was added into each well, and samples were measured at 450 nm immediately. Before testing, plasma samples were thawed and mixed by auto shaking. The measurement of TNF-α went according to the manufacturer‘s recommendation. The standards provided for each ELISA kit were used to plot each standard curve according to the manufacturer’s instructions. Absorbance readings were performed at 450 nm.

***Statistical analysis.*** Data analysis was performed using the IBM SPSS Statistics^®^ software program (Statistical Package for Social Sciences 20 for Windows). The averaged experimental results are expressed as means ± standard deviation (SD). The Mann–Whitney U test and Tukey’s test were used to evaluate the data. The results were compared between groups (CCLR-P1 and CCLR-P2; CCLR-C1 and CCLR-C2; PL-II1 and PL-II2; PL-III1 and PL-III2; CCLR-C1, CCLR-P1 and CCLR-C2, CCLR-P2; PL-II1, PL-III1 and PL-II2, PL-III2) at d0, d14, and d28 and from baseline at d0, d14, and d28. The level of significance was set at *p* < 0.05. 

## 3. Results

The age of the dogs in the study was 2.7 years ± 1.1 years (range: 1.1–5.1 years). Animals in the CCLR group were twelve mixed-breed dogs (seven females, five males), and their mean weight was 18.2 ± 7.5 kg, and mean age was 3.3 ± 1.1 years. Animals in the PL group were ten mixed-breed dogs (four females, six males), and their mean weight was 5.5 ± 1.7 kg, and mean age was 2.5 ± 1 years. Duration of lameness ranged from five to twenty-one days. 

### 3.1. TNF-α Concentration

The results of TNF-α concentration indicates a significant difference between groups of CCLR-P1 (*n* = 3) and CCLR-P2 (*n* = 3) on d14 (*p* = 0.006) (Figure 1). Results of group CCLR-P1 on d14 are decreased, while results of group CCLR-P2 on d14 are increased. When comparing TNF-α concentration between all CCLR cases treated with NSAIDs and treated with PRP, there was a significant difference between those groups on d14 (*p* = 0.001). The results of TNF-α concentration indicates a significant difference between groups PL-III1 and PL-III2 on d28 (*p* = 0.036) (Figure 2). Results of group PL-III1 indicate growth of TNF-α concentration, while at the same d28, results of group PL-III2 indicate decreased levels of cytokine.

*CCLR subgroups.* There was a significant difference in TNF-α concentrations in the group CCLR-P2 on d0 and d28 (*p* = 0.029) and on d14 and d28 (*p* = 0.016) and in the group CCLR-C2 on d0 and d28 (*p* = 0.01) and on d14 and d28 (*p* = 0.041).These results indicate increment in TNF-α concentrations up to d14, while at d28 the results shows a significant reduction in this parameter. 

*PL subgroups.* Statistical analysis indicates a significant difference in TNF-α concentration in the group PL-II2 on d0 and d28 (*p* = 0.041) and on d14 and d28 (*p* = 0.025). The results revealed increment in TNF-α levels in this subgroup up to d14 and then drastic reduction in cytokine levels. The results also indicate a significant difference in TNF-α concentration in the group PL-III1 on d0 and d14 (*p* = 0.02), revealing a reduction in TNF-α levels on d14.

### 3.2. Degree of Lameness

Lameness score did not differ at any time between CCLR-P and CCLR-C groups (*p* > 0.05) and between PL-III1 and PL-III2 groups (*p* > 0.05).

*CCLR subgroups.* The results of Tukey’s test revealed that there was a significant difference in lameness score in the group CCLR-P1 on d0 and d14 (*p* = 0.017), in the group CCLR-P2 on d0 and d14 (*p* = 0.001). Results of both groups showed reduction in lameness scores on d14. The results showed a significant difference in lameness score in the group CCLR-C2 on d0 and d14 (*p* = 0.02) and on d14 and d28 (*p* = 0.012). These results indicate a continuous reduction in lameness scores up to d28. The data are summarized in Table 1.

*PL subgroups.* The results of Tukey’s test revealed that there was a significant difference in lameness score in the group PL-II2 on d0 and d28 (*p* = 0.046). The results showed a drastic reduction in lameness degree at d14. Statistical analysis revealed a significant difference in group PL-III1 on d0 and d14 (*p* = 0.03) and d0 and d28 (*p* = 0.03). It indicates reduction in lameness scores up to d14 and maintenance of that score up to d2. Statistical analysis revealed a significant difference in group PL-III2 on d0 and d14 (*p* = 0.013) and d0 and d28 (*p* = 0.001), that indicates a continuous reduction in lameness degree up to d28. The data are summarized in Table 2.

### 3.3. Painfulness of Manipulations

Painfulness of manipulations did not differ at any time between CCLR-P and CCLR-C groups (*p* > 0.05) and between PL-III1 and PL-III2 groups (*p* > 0.05).

*CCLR subgroups.* The results of the painfulness of manipulations revealed a significant difference in results of the group CCLR-P1 on d0 and d14 (*p* = 0.001) and on d0 and d28 (*p* = 0.006), revealing reduction in painfulness up to d14 and, then, an increment up to d28. Statistical analysis showed a significant difference in the group CCLR-C1 on d0 and d14 (*p* = 0.02). These results indicate a reduction in painfulness up to d14. Statistical analysis indicates a significant difference in painfulness of manipulations in the group CCLR-P2 between d0 and d14 (*p* = 0.005) and on d0 and d28 (*p* = 0.002), and in the group CCLR-C2 on d0 and d14 (*p* = 0.005) and on d0 and d28 (*p* = 0.001). Results of both groups indicate a continuous reduction in painfulness of manipulations up to d28. The data are summarized in Table 3.

*PL subgroups.* The results of the painfulness of manipulations revealed a significant difference in the group PL-III1 on d0 and d14 (*p* = 0.013) and on d14 and d28 (*p* = 0.028). These results showed a reduction in painfulness up to d14 and, then, an increment up to d28. There was a significant difference in the group PL-III2 on d0 and d28 (*p* = 0.024) that indicates a significant decrement in painfulness of manipulations at d28. The data are summarized in Table 4.

### 3.4. Goniometry in Flexion

Goniometry in flexion did not differ at any time between CCLR-P and CCLR-C groups (*p* > 0.05). The results of statistical analysis of goniometry in flexion revealed a significant difference between groups PL-II1 and PL-II2 on d0 (*p* = 0.03), and between groups of PL-III1 and PL-III2 on d28 (*p* = 0.003) (Figure 3). When comparing the goniometry in flexion between all PL cases treated with NSAIDs and treated with PRP, there was a significant difference between those groups on d28 (*p* = 0.02).

*CCLR subgroups.* The results of statistical analysis of goniometry in flexion revealed a significant difference in the group CCLR-P2 on d0 and d28 (*p* = 0.007). These results revealed an increment in goniometry in flexion in the subgroup at d28. Statistical analysis indicates significant difference in the group CCLR-C1 on d0 and d28 (*p* = 0.039) and in the group CCLR-C2 on d0 and d28 (*p* = 0.006). The results revealed an increment in goniometry in flexion in both subgroups up to d28. 

*PL subgroups.* The results of statistical analysis of goniometry in flexion revealed a significant difference between groups PL-II1 and PL-II2 on d0 (*p* = 0.03), and between groups of PL-III1 and PL-III2 on d28 (*p* = 0.003). The results indicate an increment in goniometry in flexion in the subgroup of PL-III1 at d28 and a reduction in the same parameter in the subgroup of PL-III2 at d28 (Figure 2). There was a significant difference in goniometry in flexion in the group of PL-II2 on d0 and d14 (*p* = 0.048), on d0 and d28 (*p* = 0.007), and on d14 and d28 (*p* = 0.048). These results of this subgroup revealed a continuous decrement in goniometry in flexion up to d28. 

### 3.5. Goniometry in Extension

Goniometry in extension did not differ at any time between CCLR-P and CCLR-C groups (*p* > 0.05) and between PL-III1 and PL-III2 groups (*p* > 0.05).

*CCLR subgroups.* The results of goniometry in extension indicates a significant difference in the group CCLR-P2 on d0 and d28 (*p* = 0.05), d14 and d28 (*p* = 0.043), indicating a continuous decrement in goniometry in flexion up to d28. Similar results were observed in group CCLR-C1 on d0 and d28 (*p* = 0.034) and in group CCLR-C2 on d0 and d14 (*p* = 0.044) and d0 and d28 (*p* = 0.003). The data are summarized in Table 5.

*PL subgroups.* The results of goniometry in extension indicates a significant difference in the group PL-II2 on d0 and d28 (*p* = 0.012), showing a continuous decrement in goniometry in flexion up to d28. The data are summarized in Table 6.

### 3.6. Muscle Strength

There was a significant difference in muscle strength between group CCLR-P1 and group CCLR-P2 on d28 (*p* = 0.007), indicating an increment in muscle strength in group CCLR-P1 up to d14 and reduction in it up to d28, while the muscle strength of group CCLR-P2 was continuously increasing up to d28 (Figure 4). When comparing the muscle strength between all CCLR cases treated with NSAIDs and treated with PRP, there was a significant difference between those groups on d28 (*p* = 0.007). Muscle strength did not differ at any time between PL-III1 and PL-III2 groups (*p* > 0.05). 

*CCLR subgroups.* Statistical analysis indicates a significant difference in muscle strength in the group CCLR-P1 on d0 and d14 (*p* = 0.031), showing an increment in tested parameter up to d14. The results of muscle strength indicate a significant difference in the group of CCLR-P2 on d0 and d28 (*p* = 0.024), revealing a continuous increment in muscle strength within the subgroup up to d28. The results of Tukey’s test revealed that there was a significant difference in muscle strength in the group CCLR-C1 on d0 and d14 (I = 0.041), revealing a drastic increment in muscle strength up to d14. A significant difference was revealed in the group CCLR-C2 on d0 and d14 (*p* = 0.02) and on d0 and d28 (*p* = 0.002), showing continuous increment in muscle strength up to d28. 

*PL subgroups.* Statistical analysis indicates a significant difference in muscle strength in the group PL-III1 on d0 and d14 (*p* = 0.012) and d0 and d28 (*p* = 0.024), revealing a continuous increment in muscle strength within a subgroup up to d28.

## 4. Discussion

Canine CCLR and PL are pathologies leading to inflammation and OA in the stifle joint [8,14]. PRP has been shown as a promising treatment option for canine OA [8]. TNF-α is a cytokine participating and indicating inflammation processes [22,23,24,40]. Intra-articular injections of PRP are associated with the beneficial clinical outcome due to growth factors that can reduce local inflammation and its clinical signs [40,41]. There is a lack of research and scientific evidence of success and comparison between clinical outcomes of single intra-articular PRP injection usage and conservative treatment options of CCLR and PL in dogs. The present study aimed to evaluate and compare the use of oral NSAIDs and single intra-articular injection of PRP as CCLR and PL diseases-modifying therapy through analysis of inflammation cytokine TNF-α concentration and clinical outcomes measurement changes over a twenty-eight day study period and to compareand evaluate effectiveness and outcomes of those two different CCLR and PL management protocols. The results of the study supports the hypothesis of the sustained benefits lasting up to 28 after a single intra-articular PRP injection versus non-surgical management of PL and CCLR by using oral NSAIDs up to 14 days. To our knowledge, this is the first study providing insights into efficacy of single intra-articular PRP usage associated with oral NSAIDs treatment in cases of canine CCLR and PL. 

This study demonstrates significant changes in serum TNF-α concentration in cases of canine CCLR and PL treated by single intra-articular PRP injection. Different studies on animal models already showed reduction in TNF-α levels after PRP injections [42,43,44,45], but, to our knowledge, this is the first study comparing effects of PRP injection and usage of NSAIDs in canine CCLR and PL management. In contrast to the results reported by Arican et al. [46], this study showed a significant change in serum TNF-α level of patients treated with single intra-articular PRP injection. In this study we found a significant difference in serum TNF-α levels between patients treated by single intra-articular PRP injection and with those treated by NSAIDs at d14. In our study, we found that serum TNF-α levels in cases of canine CCLR and PL, treated by oral NSAIDs administration or single intra-articular PRP injection, indicate difference in action. In cases, treated by PRP injection, cytokines levels were higher than initial ones at d14, while in the NSAIDs group, TNF-α levels on d14 were decreased, but at d28, PRP group’s TNF-α levels decreased drastically, while NSAIDs group’s increased. Results of this study indicate that NSAIDs action is short-term while PRP effect on TNF-α levels acts as long-term modality. The results revealed distinct actions of the two treatment modalities: the cytokine levels after single PRP injection are increased on d14 possibly due to inflammation process, while after NSAIDs use, the levels are decreased on d14, but on d28, PRP still acts in manner that decreases pro-inflammatory cytokine levels, while the effect of NSAIDs ends. 

In this study, clinical outcomes were obtained and noted for all the dogs (*n* = 22) by the lameness scoring, painfulness to manipulations scoring, goniometry in flexion and extension measuring, and muscle strength scoring for the clinical effects of NSAIDs and PRP usage. These evaluations have been used in previous studies and have been accepted [38,40]. The results from our study indicate reduced and lasting lameness scores both in cases of CCLR and PL, treated by PRP injection. Patients, treated with PRP in cases of joint defects, have greater improvement in lameness scores [47]. Results of our study also indicates decreased painfulness of patients treated with PRP in both cases of CCLR, and PL. Although our results showed significant reduction in painfulness in cases of CCLR that was greater for those treated with PRP than NSAIDs. Wanstrath et al. [48] found that single PRP injection decreases pain scores in cases of canine OA significantly. Our results indicate that after using PRP, regression of lameness and painfulness is longer-term compared with the usage of NSAIDs and lasted up to 28 days while the usage of NSAIDs did not provide a lasting effect up to 28 days. 

There is lack of scientific evidence on PRP effects on joints goniometry and muscles strength. To our knowledge, this is the first research evaluating mentioned parameters. The results indicate significant changes in period of 4 weeks in goniometry (flexion and extension) of canine stifle joint in cases of CCLR (partial and complete) and PL (grade II and grade III), treated by PRP or NSAIDs. The study results showed that in cases where stifle pathologies were treated by PRP, the positive effect on goniometry and muscle strength was seen up to 28 days, whereas in cases treated with NSAIDs, the effect at day 28 of research was slightly reduced. The results of clinical outcomes are similar to what has been reported in human patients, where a single intra-articular PRP injection into the knee joint increases physical function [40]. This study has several limitations related to the lack of data about the length of continued lameness before the initial visit, tracking and recording minor variations in treatment schemes, and tracking of animal activity during the research period. The missing data would have provided a better understanding of the mechanisms of action and outcomes of a single intra-articular PRP injection compared with the usage of oral NSAIDs in treating cases of canine CCLR and PL. Furthermore, due to our biomedical study’s nature and ethical concerns, it was impossible to use a placebo group as a control group.

Differentiation in CCLR subgroups should be conducted using special diagnostic methods (such as MRI or ultrasound) and not only using orthopedic examination. These methods would be beneficial for more representable results.

Larger sample sizes, frequent follow-up periods, and study randomization would benefit further studies. However, we believe that the results of this study may be helpful for further understanding of managing canine orthopedic diseases using new treatment methods.

## 5. Conclusions

The single intra-articular platelet-rich-plasma (PRP) injection as treatment method of canine cranial cruciate ligament rupture (CCLR) and canine patellar luxation (PL) is novel and has not been studied extensively. Our study suggests that this method may be a successful treatment in terms of clinical outcomes, with longer duration of action and management of inflammation in the affected joint, contrary to oral NSAIDs. 

We conclude that a single intra-articular injection of PRP has a more beneficial effect on the inflammatory process and clinical outcomes and has a longer duration of action than oral NSAIDs in the treatment of canine CCRL or PL.

## Figures and Tables

**Figure 1 vetsci-10-00555-f001:**
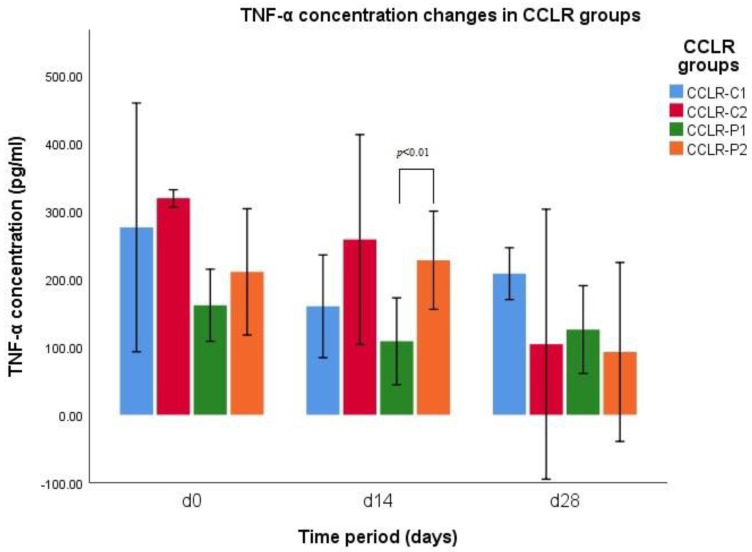
TNF−α concentration changes during different time periods (d0, d14, and d28) in CCLR−P and CCLR−C groups.

**Figure 2 vetsci-10-00555-f002:**
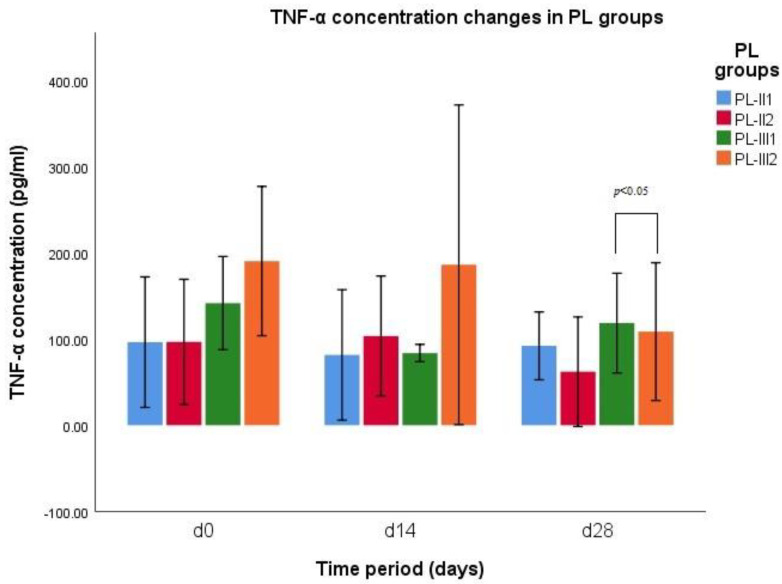
TNF−α concentration changes during different time periods (d0, d14, and d28) in PL−II and PL−III groups.

**Figure 3 vetsci-10-00555-f003:**
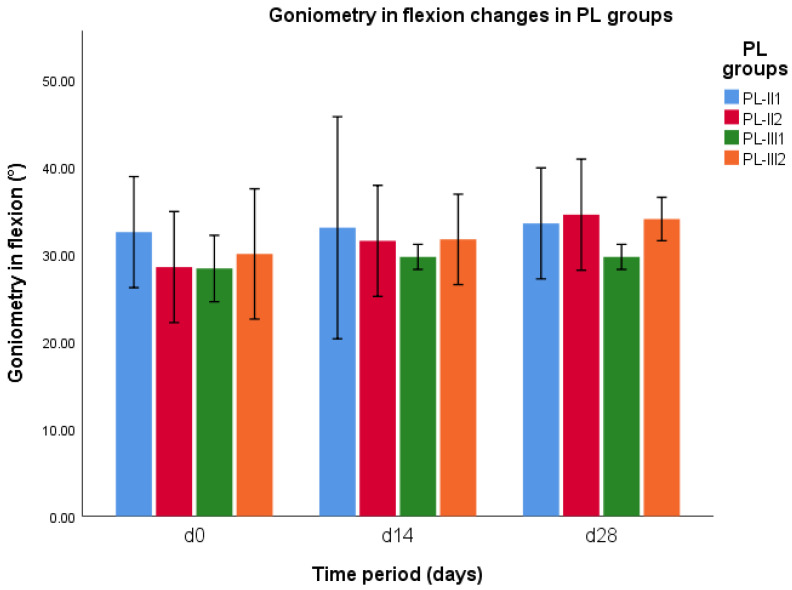
Goniometry in flexion changes during different time periods (d0, d14, and d28) in PL-II and PL-III groups.

**Figure 4 vetsci-10-00555-f004:**
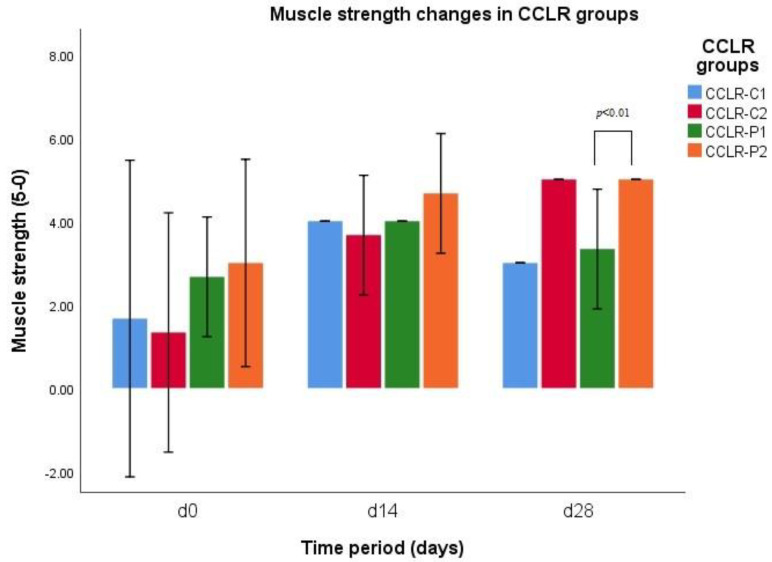
Muscle strength changes during different time periods (d0, d14, and d28) in CCLR−P and CCLR−C groups.

**Table 1 vetsci-10-00555-t001:** The association between degrees of lameness in the CCLR groups. The values present as means ± standard deviation.

Parameter	CCLR Groups	d0	d14	d28
Degree of lameness (0–5)	CCLR-P1	3.33 ± 0.58 ^a^	0.67 ± 0.58 ^a^	1.33 ± 1.15
	CCLR-P2	3.33 ± 0.58 ^b^	0.67 ± 0.58 ^b^	0.00
	CCLR-C1	3.00 ± 1.00	2.00 ± 0.00	2.00 ± 0.31
	CCLR-C2	4.33 ± 0.58 ^c^	2.00 ± 0.00 ^c^	0.33 ± 0.58 ^c^

^a^—changes in degrees of lameness of CCLR-P1 and CCLR-P2 on d0 and d14 *(p* < 0.05); ^b^—changes in degrees of lameness of CCLR-P2 on d0 and d14 (*p* < 0.01); ^c^—changes in degrees of lameness of CCLR-C2 on d14 (*p* < 0.05).

**Table 2 vetsci-10-00555-t002:** The association between degrees of lameness in the PL groups. The values present as means ± standard deviation.

Parameter	PL Groups	d0	d14	d28
Degree of lameness (0–5)	PL-II1	2.50 ± 0.7	0.50 ± 0.71	0.50 ± 0.71
	PL-II2	2.50 ± 0.7 ^a^	0.50 ± 0.71	0.00 ^a^
	PL-III1	3.67 ± 0.58 ^b,c^	1.33 ± 0.58 ^b^	1.33 ± 1.15 ^b,c^
	PL-III2	3.67 ± 0.58 ^d,e^	1.67 ± 0.58 ^d^	0.33 ± 0.58 ^d,e^

^a^—changes in degrees of lameness of PL-II2 on d0 and d28 *(p* < 0.05); ^b^—changes in degrees of lameness of PL-III1 on d0 and d14 (*p* < 0.05); ^c^—changes in degrees of lameness of PL-III1 on d0 and d28 (*p* < 0.05); ^d^—changes in degrees of lameness of PL-III2 on d0 and d14 (*p* < 0.05); ^e^—changes in degrees of lameness of PL-III2 on d0 and d28 (*p* < 0.05).

**Table 3 vetsci-10-00555-t003:** The association between painfulness of manipulations in the CCLR groups. The values present as means ± standard deviation.

Parameter	CCLR Groups	d0	d14	d28
Painfulness of manipulations (1–3)	CCLR-P1	2.00 ± 0.00 ^a,b^	0.00 ^a^	0.67 ± 0.58 ^b^
	CCLR-P2	2.33 ± 0.58 ^c,d^	0.33 ± 0.58 ^c^	0.00 ^d^
	CCLR-C1	2.67 ± 0.58 ^e^	0.33 ±0.58 ^e^	1.00 ± 1.00
	CCLR-C2	2.67 ± 0.58 ^f,g^	0.67 ± 0.58 ^f^	0.00 ^g^

^a^—changes in painfulness of manipulations of CCLR-P1 on d0 and d14 *(p* < 0.01); ^b^—changes in painfulness of manipulations of CCLR-P1 d0 and d28 (*p* < 0.01); ^c^—changes in painfulness of manipulations of CCLR-P2 on d0 and d14 (*p* < 0.01); ^d^—changes in painfulness of manipulations of CCLR-P2 on d0 and d28 (*p* < 0.01); ^e^—changes in painfulness of manipulations of CCLR-C1 on d0 and d14 (*p* < 0.05); ^f^—changes in painfulness of manipulations of CCLR-C2 on d0 and d14 (*p* < 0.01); ^g^—changes in painfulness of manipulations of CCLR-C2 on d0 and d28 (*p* < 0.01).

**Table 4 vetsci-10-00555-t004:** The association between painfulness of manipulations in the PL groups. The values present as means ± standard deviation.

Parameter	PL Groups	d0	d14	d28
Painfulness of manipulations (1–3)	PL-II1	1.00 ± 0.00	0.50 ± 0.78	0.00
	PL-II2	1.00 ± 0.00	0.00	0.00
	PL-III1	2.33 ± 0.58 ^a,b^	0.33 ±0.58 ^a^	0.67 ± 0.58 ^b^
	PL-III2	2.00 ± 1.00 ^c^	0.67 ± 0.58	0.00 ^c^

^a^—changes in painfulness of manipulations of PL-III1 on d0 and d14 *(p* < 0.05); ^b^—changes in painfulness of manipulations of PL-III1 on d14 and d28 (*p* < 0.05); ^c^—changes in painfulness of manipulations of PL-III2 on d0 and d28 (*p* < 0.05).

**Table 5 vetsci-10-00555-t005:** The association between goniometry in extension in the CCLR groups. The values present as means ± standard deviation.

Parameter	CCLR Groups	d0	d14	d28
Goniometry in extension (°)	CCLR-P1	149.00 ± 1.00	150.67 ± 1.15	151.33 ± 1.53
	CCLR-P2	146.00 ± 1.73	148.67 ± 1.53 ^a^	153.00 ± 2.00 ^a^
	CCLR-C1	146.67 ± 1.53 ^b^	149.00 ± 1.00	150.00 ± 1.00 ^b^
	CCLR-C2	144.67 ± 1.15 ^c,d^	148.33 ± 1.53 ^d^	151.33 ± 1.53 ^d^

^a^—changes in goniometry in extension of CCLR-P2 on d14 and d28 *(p* < 0.05); ^b^—changes in goniometry in extension of CCLR-C1 on d0 and d28 (*p* < 0.05); ^c^—changes in goniometry in extension of CCLR-C2 on d0 and d14 (*p* < 0.05); ^d^—changes in goniometry in extension of CCLR-C2 on d0 and d28 (*p* < 0.01).

**Table 6 vetsci-10-00555-t006:** The association between goniometry in extension in the PL groups. The values present as means ± standard deviation.

Parameter	PL Groups	d0	d14	d28
Goniometry in extension (°)	PL-II1	147.50 ± 2.12	148.50 ± 2.12	149.00 ± 2.83
	PL-II2	149.50 ± 0.71 ^a^	151.50 ± 0.71	154.50 ± 0.71 ^a^
	PL-III1	145.67 ± 3.79	146.67 ± 3.06	147.33 ± 3.21
	PL-III2	147.33 ± 2.52	148.33 ± 1.53	150.67 ± 1.53

^a^—changes in goniometry in extension of PL-II2 on d0 and d28 *(p* < 0.01).

## Data Availability

The data presented in this study are available upon request from the corresponding author.

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
