# Peer review of "The Single Intra-Articular Injection of Platelet-Rich Plasma vs. Non-Steroidal Anti-Inflammatory Drugs as Treatment Options for Canine Cruciate Ligament Rupture and Patellar Luxation"

_vetsci, 2023, doi:10.3390/vetsci10090555_

Round 1

Reviewer 1 Report

Thank you for the opportunity to review your manuscript in the topic of platelet products where further research is needed on its efficacy in our clinical canine patients. I have provided comments below which I think will contribute to improving the clarity and quality of the manuscript and its potential impact:

Line 41-43: It is unclear what was found to be signification – these values compared to baseline within groups, or between groups compared at the same time point, or both? 

Line 61: Defining PL as medial displacement of the tibial tuberosity is incorrect for patellar luxation as a whole; this only describes medial luxations and may not be present even in all cases of medial PL. There are also a number of other contributing factors to PL than just the trochlear groove and tibial tuberosity, other skeletal deformities contribut 

Line 63-65: These sentences are awkwardly short and could be expanded upon to provide additional details (e.g. PL can be medial, lateral, or bilateral with medial being most common, brief description of the four grades of PL) 

Line 83: “Cytokines and growth effectively palliate...” missing the word “factors” in this sentence 

Line 92: Please provide a hypothesis statement. 

Line 109: “Only patients with PL grade II and III were included in the study...” It would be helpful to clarify here that you are speaking of the subset of dogs with PL, only grade II and III were included. Otherwise this is slightly confusing, since this sentence sounds contradictory to the preceding sentence 

Line 121: Please provide clarification on how partial vs complete tears were determined (suspicion based on exam?) 

Line 139-143: What is this control group for? A more informative control group would be a group of similarly affected dogs who receive placebo treatment (oral/IA). Otherwise I am not sure what the goal of using normal healthy dogs is in this study. 

Line 144: Typo “CCCLR” 

Line 145: CX was given for 10 days but glucosamine/chondroitin supplement given for 14 days? This contradicts the abstract which says both were given for 14 days. Please rectify. 

Line 146: Please provide dose information on the glucosamine/chondroitin supplement used 

Line 144-149: It would be helpful in this paragraph to clarify which groups received each treatment. You use the term “medically treated” which could be confusing, since both oral medications and IA injections could be considered a medical treatment compared to surgical treatment.  

Line 159: Please describe how limb circumference was performed (specify thigh circumference, standing vs recumbent, level on limb in which measurement was performed). Also, how was this data handled since limb circumference will vary with size of the dog? 

Line 160-161: Please provide details of the Canine Muscle Strength Test used. Has the particular test used been previously described in the literature? If so, please provide a citation.  

Line 163: Was this veterinarian blinded to the treatment groups? 

Line 170: Please specify that only the CCLR-P2, CCLR-C2, PL-II2 and PL-III2 group dogs underwent sedation 

Line 201: Please provide additional statistical information regarding variables considered in the statistical analysis – did you compare within groups from baseline to each time point, or between groups at given time point, or both? This should be clarified in the stats section here rather than in the Results section.  

Line 206: Please indicate how many dogs were in each subgroup.  

Line 212: Minor typo in the reported p-value, a comma was used instead of a period “(p=0,029)” 

General comment about results delivery:  

-Consider presenting the data for each outcome measure first discussing the larger groups before splitting into the subsets. I think this would improve clarity of the results delivery and be easier to read. It would also point out clinical relevance which could be expanded upon in the discussion; e.g. does PRP have a more profound impact in partial vs complete CCLR or grade II vs grade III PL? I think this would also be more appropriate since your treatment groups are so small, to highlight the data as a whole before breaking into the subgroups. 

-It would be helpful to tell the reader what significant differences were seen – increased values or decreased values – so they can create a general clinical idea of positive vs negative response to treatment. It would also be helpful to clearly indicate differences between groups versus over time.  

-Graphical representation of the data would be very helpful as a visual aid. 

Table 1 & 2: I'm unclear what you mean by girthometry reported in degrees; either this is incorrectly reported or I am unfamiliar with how you obtained and are reporting these numbers. Please provide clarification. 

Table 2: Goniometry in extension PL-II1 d0 I think you have a typo here, 14.50 degrees extension seems wrong! 

Table 2: Girthometry PL-II1 d14 what happened that it went from 54.15 to 30.60 to 56.55?

Line 287: “...cytokine that participates in inflammation processes and indicates it.” Awkward sentence structure here ending the sentence in “it”. Please edit to improve clarity of this sentence.

Line 288: Minor grammar error here, should be “Intra-articular injections of PRP are associated...” 

Line 321: n=32 is incorrect here, you only had 22 dogs in treatment groups in which you were assessing clinical outcomes. The 10 “control dogs” were never mentioned beyond the Materials and Methods section and I'm not sure why they were included.

Line 327: Typo “treaded” 

Line 333: This last sentence here is misleading. Your results demonstrate reduced lameness and pain up to day 28 (4 weeks), while short-term NSAID use did not provide a lasting improvement up to day 28.  

Line 341: Again, you performed final outcome measurements at day 28, so it is misleading to say a positive effect was “seen even after 4 weeks” 

Line 343: Please clarify that you mean “human patients” rather than “medical patients” 

Line 345-346: I’m not sure how the data reported here are similar to this prior study cited here, please provide clarification. 

Additional comments for discussion:  

-I would recommend further discussing possible limitations of your study including the small sample size for each subset of treatment groups, only evaluating up to 28 days, unblinded, non-randomized study design, lack of a placebo group.

Minor grammatical errors and awkward sentences, I have tried to point these out in the general comments section so they can be corrected.

Author Response

            We appreciate your valuable comments and advices, and like to thank you for your time and important contribution to improving our manuscript. Please find our responses to each of your comments:

  • Line 41-43: It is unclear what was found to be signification – these values compared to baseline within groups, or between groups compared at the same time point, or both?

Our answer: these values were compared between groups at the same time point. To clarify this at our manuscript, we changed our statement as following: “Results indicated that there was a significant difference in serum TNF-α levels in partial CCLR group on day 14 (p<0.01), between all CCLR cases on day 14 (p<0.01), and in PL III group on day 28 (p<0.05), comparing results between the subgroups at the same time point and within subgroups from baseline. Results indicates a significant difference in goniometry in flexion between groups of PL II and PL I, II on day 28 (p<0.01).”

  • Line 61: Defining PL as medial displacement of the tibial tuberosity is incorrect for patellar luxation as a whole; this only describes medial luxations and may not be present even in all cases of medial PL. There are also a number of other contributing factors to PL than just the trochlear groove and tibial tuberosity, other skeletal deformities contribute.

Our answer: we have modified this definition as follows: “Patellar luxation (PL) is one of the most common orthopedic diseases that can be con-genital or traumatic in its origin. Congenital PL is resulting from the quadriceps mechanism malalignment and inadequate trochlear groove that leads to the development of degenerative joint disease, painfulness, secondary OA and osteoarthritic changes in canine patients [12-15].”

  • Line 63-65: These sentences are awkwardly short and could be expanded upon to provide additional details (e.g. PL can be medial, lateral, or bilateral with medial being most common, brief description of the four grades of PL).

Our answer: the sentences were changed as follows: “PL can be medial, lateral, or bidirectional, medial being the vast majority of luxations [16-21]. PL in canine is graded into four grades [16]. Grade I – patella does not spontaneously luxate, but can be manually luxated, but afterwards comes back into trochlear groove. Grade II – the patella may be manually displaced or may luxate with flexion/extension of the stifle; the patella remains luxated until manually reduced. Grade III – the patella remains luxated majority of the time but may be manually reduced; flexion/extension of stifle results in relaxation. Grade IV – the patella is permanently luxated and cannot be manually reduced.”

  • Line 83: “Cytokines and growth effectively palliate...” missing the word “factors” in this sentence.

Our answer: the sentence has been changed to: “Cytokines and growth factors effectively palliate inflammation and initiate the anabolic process and tissue regeneration [37].”

  • Line 92: Please provide a hypothesis statement.

Our answer: our hypothesis statement was written as  “Our hypothesis was that a single intra-articular PRP injection affects TNF-α levels and clinical outcomes in canine CCLR (partial and complete) and PL (grade II and III) as much or more than oral NSAID administration and is an effective treatment option for giving pathologies.”

  • Line 109: “Only patients with PL grade II and III were included in the study...” It would be helpful to clarify here that you are speaking of the subset of dogs with PL, only grade II and III were included. Otherwise this is slightly confusing, since this sentence sounds contradictory to the preceding sentence

Our answer: we are thankful for your suggestion for changes and we agree with them. The statement was clarified as “Only patients with PL grades II and III were included in the study from all PL cases, as owners of dogs with PL grade I refused treatment, and all owners of patients with PL grade IV required recommended surgical treatment.”

  • Line 121: Please provide clarification on how partial vs complete tears were determined (suspicion based on exam?)

Our answer: partial vs complete tears were determined by difference in orthopedic examination: if animal shown positive drawer test but negative tibial compression test, it was determined as having partial tear, while if animal shown positive tibial compression test, it was determined as having complete tear. We understand that this determination needs to be improved by MRI scanning, arthroscopy or surgical revision, but due to nature of our study we chose this determination method. The statement was clarified as “The animals under study were twelve mixed breed-dogs (seven females, five males, mean weight 18.2 ± 7.5 kg, mean age 3.3 ± 1.1 years) diagnosed with CCLR due to the clinical (lameness, painfulness of manipulations, positive stifle drawer and tibial compression tests) and radiological (signs of joint effusion, OA changes, changes in tibial plateau angle) examinations findings. Partial and complete CCLR were determined by difference in orthopedic examination: if animal shown positive drawer test but negative tibial compression test, it was determined as having partial tear, while if animal shown positive tibial compression test, it was determined as having complete tear.”

  • Line 139-143: What is this control group for? A more informative control group would be a group of similarly affected dogs who receive placebo treatment (oral/IA). Otherwise I am not sure what the goal of using normal healthy dogs is in this study.

Our answer: we are thankful for your comments and suggestions. Due to our research nature (our research being a clinical research and not an experimental study with laboratory animals), we were unable to have a placebo group, because it would be unethical to use placebo treatment in client-owned dogs. We agree with your comment that the usage of our designed control group have no greater goal in this research and should be executed.

  • Line 144: Typo “CCCLR”

Our answer: it was changed an written as “CCLR”.

  • Line 145: CX was given for 10 days but glucosamine/chondroitin supplement given for 14 days? This contradicts the abstract which says both were given for 14 days. Please rectify.

Our answer: it was a writing mistake and we changed it as “Each dog of group CCLR-P1, CCLR-C1, PL-II1, and PL-III1 received CX (2 mg/kg, orally every twenty-four hours for fourteen days; Vétoquinol SA, France) and supplements of glucosamine/chondroitin (,2 µg cyanocobalamin, 53,1 µg folic acid, 2,6 mg thamin, 0,51 mg riboflavin, 1,68 mg pyridoxine, 2,26 mg niacin, 0,71 mg vitamin E, 55,5 mg calcium, 44,6 mg phosphorus P, 52,5 mg omeha 3 nutritients, 225 mg glucosamine sulphate 2KCl, 162 mg chondroitin sulphate, 59,94 mg MSM per 22-33 kg due to manufacturer’s recommendation, orally every twenty-four hours for fourteen days; Bob Martin, UK).”

  • Line 146: Please provide dose information on the glucosamine/chondroitin supplement used.

Our answer: dose information of manufacturer’s recommendations is written as: “7.2 µg cyanocobalamin, 53.1 µg folic acid, 2.6 mg thamin, 0.51 mg riboflavin, 1.68 mg pyridoxine, 2.26 mg niacin, 0.71 mg vitamin E, 55.5 mg calcium, 44.6 mg phosphorus P, 52.5 mg omega 3 nutritients, 225 mg glucosamine sulphate 2KCl, 162 mg chondroitin sulphate, 59.94 mg MSM per 22-33 kg due to manufacturer’s recommendation, orally every twenty-four hours for fourteen days; Bob Martin, UK.”

  • Line 144-149: It would be helpful in this paragraph to clarify which groups received each treatment. You use the term “medically treated” which could be confusing, since both oral medications and IA injections could be considered a medical treatment compared to surgical treatment. 

Our answer: we agree with your remark, and the information has been corrected and presented as follows: “Each dog of group CCLR-P1, CCLR-C1, PL-II1, and PL-III1 received CX (2 mg/kg, orally every twenty-four hours for fourteen days; Vétoquinol SA, France) and supplements of glucosamine/chondroitin (7.2 µg cyanocobalamin, 53.1 µg folic acid, 2,6 mg thamin, 0.51 mg riboflavin, 1.68 mg pyridoxine, 2.26 mg niacin, 0.71 mg vitamin E, 55.5 mg calcium, 44.6 mg phosphorus P, 52.5 mg omega 3 nutritients, 225 mg glucosamine sulphate 2KCl, 162 mg chondroitin sulphate, 59.94 mg MSM per 22-33 kg due to manufacturer’s recommendation, orally every twenty-four hours for fourteen days; Bob Martin, UK). Each group of dogs that were treated with PRP injections received a single intra-articular PRP injection.”

  • Line 159: Please describe how limb circumference was performed (specify thigh circumference, standing vs recumbent, level on limb in which measurement was performed). Also, how was this data handled since limb circumference will vary with size of the dog?

Our answer: Muscle mass was estimated by measuring the limb circumference with a Gulick tape measure while animal was standing in the midway between hip and the stifle, as it is described in literature (40,41 references). It was described in manuscription as “Muscle mass was estimated by measuring the limb circumference with a Gulick tape measure while animal was standing in the midway between hip and the stifle [40, 41].” We are aware of variation of limb circumference depending on the dog’s sizes, but in this research we evaluated changes in limb circumference of each patient individually due to given treatment, so variations between different individuals was not affecting the results.

  • Line 160-161: Please provide details of the Canine Muscle Strength Test used. Has the particular test used been previously described in the literature? If so, please provide a citation. 

Our answer: Canine Muscle Strength Tests were performed for muscle strength evaluation. Test is performed two times on standing animal: first time in normal standing position, the second time – while animal is standing with thoracic limbs on the incline. During the testing, one pelvic leg is lifted from the ground while evaluating the ability of a limb to maintain a static standing position. The muscle strength is graded into 0-5 grades. The test is modified from human patient’s testing and is described in literature (Duerr, F. Canine Lameness. John Wiley &Sons, 2019, 81-83.). It was clarifies in manuscript as “Canine Muscle Strength Tests were performed for muscle strength evaluation. Test is performed two times on standing animal: first time in normal standing position, the second time – while animal is standing with thoracic limbs on the incline. During the testing, one pelvic leg is lifted from the ground while evaluating the ability of a limb to maintain a static standing position. The muscle strength is graded into 0-5 grades [42].”

  • Line 163: Was this veterinarian blinded to the treatment groups?

Our answer: veterinarian was not blinded to the treatment groups due to nature of the study (the same veterinarian did all the procedures). It was clarified in manuscript as “All testing was performed and evaluated by the same veterinarian which was not blinded to the treatment groups.”

  • Line 170: Please specify that only the CCLR-P2, CCLR-C2, PL-II2 and PL-III2 group dogs underwent sedation.

Our answer: the specification was presented as: “Each dog of groups CCLR-P2, CCLR-C2, PL-II2, and PL-III2 was sedated with 5 mcg/kg of dexmedetomidine hydrochloride and 0.4 mg/kg of butorphanol given intramuscularly.”

  • Line 201: Please provide additional statistical information regarding variables considered in the statistical analysis – did you compare within groups from baseline to each time point, or between groups at given time point, or both? This should be clarified in the stats section here rather than in the Results section. 

Our answer: additional information was written in the Statistics section as: “The Mann-Whitney U test and Tukey’s test were used to evaluate the data. The results were compared between groups (CCLR-P1 and CCLR-P2; CCLR-C1 and CCLR-C2; PL-II1 and PL-II2; PL-III1 and PL-III3; CCLR-C1, CCLR-P1 and CCLR-C2, CCLR-P2; PL-II1, PL-III1 and PL-II2, PL-III2) at d0, d14 and d28 and within from baseline at d0, d14 and d28. The level of significance was set at p<0.05.”

  • Line 206: Please indicate how many dogs were in each subgroup. 

Our answer: according to your comment, we indicated the number of dogs in each subgroup, please see the manuscript. “The results of TNF-α concentration indicates a significant difference between groups of CCLR-P1 (n=3) and CCLR-P2 (n=3) on d14 (p=0.006).”

  • Line 212: Minor typo in the reported p-value, a comma was used instead of a period “(p=0,029)”

Our answer: thank you for your remark, it was corrected to “(p=0.029)“.

  • General comment about results delivery: 
  • -Consider presenting the data for each outcome measure first discussing the larger groups before splitting into the subsets. I think this would improve clarity of the results delivery and be easier to read. It would also point out clinical relevance which could be expanded upon in the discussion; e.g. does PRP have a more profound impact in partial vs complete CCLR or grade II vs grade III PL? I think this would also be more appropriate since your treatment groups are so small, to highlight the data as a whole before breaking into the subgroups.

Our answer: we are thankful for your remarks and suggestions and we agree with them completely. The results were rewritten due to your comments, please see the manuscript.

  • -It would be helpful to tell the reader what significant differences were seen – increased values or decreased values – so they can create a general clinical idea of positive vs negative response to treatment. It would also be helpful to clearly indicate differences between groups versus over time. 

Our answer: we are thankful for your remarks and suggestions and we agree with them completely. Clarifications about increasing/decreasing of values were added in the Results section, please see the manuscript.

  • -Graphical representation of the data would be very helpful as a visual aid.

Our answer: we are thankful for your comment and agree with it. We think that given tables are necessary for our data representation, but graphics of most important results were added, please see the manuscript.

  • Table 1 & 2: I'm unclear what you mean by girthometry reported in degrees; either this is incorrectly reported or I am unfamiliar with how you obtained and are reporting these numbers. Please provide clarification.

Our answer: we thank you for your comment and clarify that it was a typing mistake and all girthometry results were corrected to “mm”.

  • Table 2: Goniometry in extension PL-II1 d0 I think you have a typo here, 14.50 degrees extension seems wrong!

Our answer: we thank you for your comment and clarify that it was a typing mistake and the correct result is “147.50”.

  • Table 2: Girthometry PL-II1 d14 what happened that it went from 54.15 to 30.60 to 56.55?

Our answer: we thank you for your comment and clarify that it was a writing mistake and the correct result is 55.60. It was changed in the manuscript.

  • Line 287: “...cytokine that participates in inflammation processes and indicates it.” Awkward sentence structure here ending the sentence in “it”. Please edit to improve clarity of this sentence.

Our answer: the sentence was corrected and provided as “TNF-α is a cytokine participating and indicating inflammation processes [23-25,41].”

  • Line 288: Minor grammar error here, should be “Intra-articular injections of PRP are associated...”

Our answer: the grammatical error was corrected and provided as: “Intra-articular injections of PRP are associated with the beneficial clinical outcome due to growth factors that can reduce local inflammation and its clinical signs [42,43].”

  • Line 321: n=32 is incorrect here, you only had 22 dogs in treatment groups in which you were assessing clinical outcomes. The 10 “control dogs” were never mentioned beyond the Materials and Methods section and I'm not sure why they were included.

Our answer: we appreciate insight, and the information has been adjusted accordingly: “In this study, clinical outcomes were obtained and noted for all the dogs (n=22) by the lameness scoring, painfulness to manipulations scoring, goniometry in flexion and extension measuring, girthometry of stifle joint measuring, and muscle strength scoring for the clinical effects of NSAIDs and PRP usage.”

  • Line 327: Typo “treaded”

Our answer: it was corrected to “treated”.

  • Line 333: This last sentence here is misleading. Your results demonstrate reduced lameness and pain up to day 28 (4 weeks), while short-term NSAID use did not provide a lasting improvement up to day 28. 

Our answer: it was corrected and provided as: “Our results indicate that after using PRP, regression of lameness and painfulness is longer-term compared with the usage of NSAIDs and lasted up to 28 days.”

  • Line 341: Again, you performed final outcome measurements at day 28, so it is misleading to say a positive effect was “seen even after 4 weeks”

Our answer: it was changed and written as “The study results showed that in cases where stifle pathologies were treated by PRP, the positive effect on goniometry, girthometry and muscle strength was seen up until 28 days, when in cases treated with NSAIDs the effect at day 28 of research was slightly reduced.”

  • Line 343: Please clarify that you mean “human patients” rather than “medical patients”

Our answer: it was corrected and provided as: “The results of clinical outcomes are similar to what has been reported the human patients’ <…>”.

  • Line 345-346: I’m not sure how the data reported here are similar to this prior study cited here, please provide clarification.

Our answer: we agree with your comment and suggest to execute this comparison from the manuscript.

  • Additional comments for discussion: 
  • -I would recommend further discussing possible limitations of your study including the small sample size for each subset of treatment groups, only evaluating up to 28 days, unblinded, non-randomized study design, lack of a placebo group.

Our answer: we thank you for your comments and suggestions how to further discuss limitations of our study. We changed our discussion of our study limitations and written it as “A choice to use client-owned dogs can be a strength but also a limitation of our pilot study; treatment of orthopedic pathologies of clients’ owned dogs could cause some errors: variations in treatment schemes, in animals’ activity, different periods of continued lameness before the first visit. Due to study’s nature it was impossible to use a placebo group. This research has to be continued by focusing on a larger sample size of patients, and to be extended, with more frequent follow-up periods, should be randomized. However, we believe that the results of this study may be useful for understanding that other appropriate treatment methods are possible for curing canine orthopedic diseases.”

            We hope that our changes made for the manuscript will meet your requirements and fulfills them. We are open to any further discussion or improvement of this manuscript – we believe that this research could help veterinarians to better understand usage of platelets rich products, so we are eager to improve our research as much as possible.

Reviewer 2 Report

I read this article with a great interest. The manuscripte descibes  data that can be very useful for practicing veterinarians, especially in this particuler cases in which surgical treatment can no be performed. All aricle chapters were planned , prepared,  and presented logically , and interested way. Idea of the study, methodology, statisctic analysis were carried out very properly. Authors have made a thorough analysis of world references. In my opinion an article: "The single intra-articular injection of platelet-rich plasma v.s.  non-steroidal anti-inflammatory drugs as treatment options for  canine cruciate ligament rupture and patellar luxation" is worth to be publish in Veterinary Sciences.

Author Response

We appreciate your valuable comments and opinion, and like to thank you for your time and important contribution to improving our manuscript and for acceptance of our work.

Reviewer 3 Report

The Authors describe a comparison between two non surgical treatment for cruciate ligament ropture in dogs. Their results highlight the intrarticular  injection of platelet-rich plasma more useful than the use of non-steroidal antinfiammatory drugs. The manuscript is well made, the introduction a bit short but well centered, material and methods appropriate, results clear and undoubthly evident. I think that this work can be useful for dog clinicians and it  deserves to be published.   

Author Response

(The authors gave the same response as above.)

Round 2

Reviewer 1 Report

Thank you for the opportunity to review your revised manuscript. You have made significant improvements, especially to the Results section. There are some additional areas where I would suggest continued improvements or require clarification to improve reader comprehension which I have outlined below, I also pointed out some typos as I found them so that these can be corrected.

Line 17: Number agreement – “Cranial cruciate ligament rupture and patellar luxation are one of the…” the use of “one” is incorrect. Consider editing to “…are among the most…”

Line 34: Missing a word here “…single intra-articular injection of PRP treatment outcomes…” I think you need the word “on” between PRP and treatment

Line 38: “PRP injection into stifle joint.” Please clarify “into the affected stifle joint.”

Line 44-45: The sentence is not clear which groups you are discussing – “…between groups of PL II and PL I, II on day 28” Please edit.

Line 40-45: This addition to the abstract is helpful in identifying the key results. However, I would still like to see further clarification here to support the conclusion statement. You say there is a “significant difference” in serum TNF-alpha levels and goniometry, but please clarify if these were increases or decreases so the reader can better understand your findings from the abstract alone.

Line 62: There is unnecessary repetition in this sentence. “Degenerative joint disease”, “secondary OA”, and “osteoarthritic changes” all mean the same thing. Please edit.

Line 101: I think you mean “the given pathologies” instead of “giving pathologies”

Line 112: Please correct patient numbers to 22 to remove the 10 patients from the “control group” that are not part of this study.

Line 113: This sentence should be deleted since you are removing discussion of the control group.

Line138-139 and 148-149: Please indicate clearly how many dogs were in each subgroup. You provide n for the groups but this needs to be clear for the subgroups as well.

Line 153-157: Thank you for removing this.

Line 165-167: You repeat the same sentence twice, please correct.

Line 177-184: Thank you for these additions.

Line 201: Missing a word here – “…injected into the damaged stifle joint 1.5 inch…” should be “…injected into the damaged stifle joint using a 1.5 inch…” or something similar.

Line 227: Typo “PL-III3”, please correct.

Line 231-232: You list the age and weight of each group of dogs in the materials and methods, it would be more appropriate to move that down to your results section and replace these lines here since readers need to know the signalment of the different groups of dogs more than all of the dogs (especially since there is such a difference in body weight between the CCLR and PL groups).

Line 238: “Table 1” should be “Figure 1” here

Figure 1: This figure is very helpful to visualize your data. I’d like to see this figure expanded or additional figures included to provide visualization of TNF-alpha concentration for all of the subgroups (example: provide a figure for the 4 CCLR subgroups and a separate table for the 4 PL subgroups). A legend could be added to indicate where significance was noted.

Line 251: Typo “subrgoups”

Line 264-276 (lameness scores), 280-293 (pain on manipulation), 319-328 (goniometry in extension), 330-332 (girthometry), 334-352 (muscle strength): Thank you, these additions here are so helpful in understanding your data. Additional figures should be considered here as well so the reader can visualize and even better understand the level of changes (or lack of changes) seen in the data in the various groups.

Line 336: Missing a space between “CCLR-P1” and “up”

Table 1: In the CCLR groups column, there are some typos where “CCLR-C2” was mistyped as “CCLT-C2”. Please correct.

Table 1 and Table 2: Thank you for changing the units of girthometry but I think you actually mean “cm” instead of “mm”. I am still struggling with the use of this outcome measure, especially given that you did not find any meaningful results, and I would strongly urge you to remove this from the manuscript. First of all, the numbers you reported in the table don’t seem reasonable given the size of dogs you reported were in the study. For example, thigh circumference of ~75 (assuming cm not mm) would be a massive dog, especially assuming we have muscle atrophy occurring given the presence of a complete cranial cruciate tear. Additionally, thigh circumference is not a standardized measurement and is widely variable depending on the size of each individual dog, so to look at this in groups of heterogenous dogs doesn’t make sense statistically. It would be a reasonable measure to look at changes for each individual dog which would require redoing the stats on this part.

Line 372: Typo “There as…” should be “There is…”

Line 380: It is misleading to discuss the “long-term activity” of PRP in this way. From your results, it is fair to say that the study supported the hypothesis of the benefits of a single PRP injection over NSAIDs over a 28 day period. Maybe use the word “sustained” instead of “long-term”.

Line 429-436: Some English editing is needed in this added paragraph more so than throughout the rest of the manuscript.

Line 436: PRP is not “curing” canine orthopedic disease. Please edit.

Line 445. Period missing at the end of the sentence.

I have tried to point out specific grammatical issues in the general comments so that these can be easily identified and corrected. The main issue noted was missing articles.

Author Response

We appreciate your valuable comments and advices, and like to thank you for your time and important contribution to improving our manuscript. Please find our responses to each of your comments:

  • Line 17: Number agreement – “Cranial cruciate ligament rupture and patellar luxation are one of the…” the use of “one” is incorrect. Consider editing to “…are among the most…”

Our answer: the sentence is changed to: “Cranial cruciate ligament rupture and patellar luxation are among the most common orthopedic pathologies in canine patients.”

  • Line 34: Missing a word here “…single intra-articular injection of PRP treatment outcomes…” I think you need the word “on” between PRP and treatment

Our answer: the grammatical error was corrected and provided as: “The main purpose of this study was to evaluate and compare the use of oral NSAIDs and single intra-articular injection of PRP on treatment outcomes in cases of canine CCLR and PL.”

  • Line 38: “PRP injection into stifle joint.” Please clarify “into the affected stifle joint.”

Our answer: the sentence has been changed to: Dogs diagnosed with CCRL (n=12) and PL (n=10) were subgrouped by the severity of pathologies and given treatment: half of the CCRL and PL groups were orally administered NSAIDs and supplements for 14 days, and the other half received a single intra-articular PRP injection into affected stifle joint.

  • Line 44-45: The sentence is not clear which groups you are discussing – “…between groups of PL II and PL I, II on day 28” Please edit.

Our answer: we agree with your remark, and the information has been corrected and presented as follows: “Results indicated that there was a significant difference in serum TNF-α levels in partial CCLR group on day 14 (p<0.01), between all CCLR cases on day 14 (p<0.01), and in PL III group on day 28 (p<0.05), comparing results between the subgroups at the same time point and within sub-groups from baseline. Results indicates a significant difference in goniometry in flexion between groups of PL II and PL III on day 28 (p<0.01).”

  • Line 40-45: This addition to the abstract is helpful in identifying the key results. However, I would still like to see further clarification here to support the conclusion statement. You say there is a “significant difference” in serum TNF-alpha levels and goniometry, but please clarify if these were increases or decreases so the reader can better understand your findings from the abstract alone.

Our answer: we are thankful for your remarks and suggestions and we agree with them completely. This part of the manuscripts’ abstract was rewritten due to your comments as following: “The results of TNF-α concentration indicates a significant difference between groups of differently treated partial CCLR groups on d14 (p=0.006). Results of group CCLR-P1 on d14 were decreased, while results of group CCLR-P2 on d14 were increased. When comparing TNF-α concentration between all CCLR cases treated with NSAIDs and treated with PRP, there was a significant difference between those groups on d14 (p=0.001). The results of TNF-α concentration indicates a significant difference between groups of differently treated PL-III on d28 (p=0.036). Results of group PL-III1 indicates growth of TNF-α concentration, while at the same d28 results of group PL-III2 indicates decreased levels of cytokine, comparing results between the subgroups at the same time point and within subgroups from baseline. Results indicates a significant difference in muscle strength between group CCLR-P1 and group CCLR-P2 on d28 (p=0.007), indicating an increment of muscle strength in group CCLR-P1 up to d14 and reduction of it up to d28, muscle strength of group CCLR-P2 increasing up to d28. When comparing the muscle strength between all CCLR cases treated with NSAID’s and treated with PRP, there was a significant difference between those groups on d28 (p=0.007).”

  • Line 62: There is unnecessary repetition in this sentence. “Degenerative joint disease”, “secondary OA”, and “osteoarthritic changes” all mean the same thing. Please edit.

Our answer: it was corrected and provided as: “Congenital PL is resulting from the quadriceps mechanism malalignment and inadequate trochlear groove that leads to the development of degenerative joint disease, and painfulness in canine patients [12-15].”

  • Line 101: I think you mean “the given pathologies” instead of “giving pathologies”

Our answer: the grammatical error was corrected and provided as: “Our hypothesis was that a single intra-articular PRP injection affects TNF-α levels and clinical outcomes in canine CCLR (partial and complete) and PL (grade II and III) as much or more than oral NSAID administration and is an effective treatment option for the given pathologies.”

  • Line 112: Please correct patient numbers to 22 to remove the 10 patients from the “control group” that are not part of this study.

Our answer: it was corrected and provided as: “All patients (n=22) enrolled in the study were mixed-breed companion dogs.”

  • Line 113: This sentence should be deleted since you are removing discussion of the control group.

Our answer: the sentence was removed from the manuscript.

  • Line138-139 and 148-149: Please indicate clearly how many dogs were in each subgroup. You provide n for the groups but this needs to be clear for the subgroups as well.

Our answer: according to your comment, we indicated the number of dogs in each subgroup, please see the manuscript.

  • Line 165-167: You repeat the same sentence twice, please correct.

Our answer: the second sentence was removed from the manuscript.

  • Line 201: Missing a word here – “…injected into the damaged stifle joint 1.5 inch…” should be “…injected into the damaged stifle joint using a 1.5 inch…” or something similar.

Our answer: it was corrected and provided as: “After aseptic preparation, the ACP (2 mL) was injected into the damaged stifle joint using 1.5-inch 20-gauge needle.”

  • Line 227: Typo “PL-III3”, please correct.

Our answer: the grammatical error was corrected and provided as: “PL-III2”.

  • Line 231-232: You list the age and weight of each group of dogs in the materials and methods, it would be more appropriate to move that down to your results section and replace these lines here since readers need to know the signalment of the different groups of dogs more than all of the dogs (especially since there is such a difference in body weight between the CCLR and PL groups).

Our answer: we are thankful for your remarks and suggestions and we agree with them completely. Clarifications about the weight of animals in CCLR and PL groups was added to the manuscript as: “Animals in the CCLR group were twelve mixed breed-dogs (seven females, five males), their mean weight 18.2 ± 7.5 kg, mean age 3.3 ± 1.1 years. Animals in the PL group were ten mixed-breed dogs (four females, six males), their mean weight 5.5 ± 1.7 kg, mean age 2.5 ± 1 years.”

  • Line 238: “Table 1” should be “Figure 1” here

Our answer: it was corrected to: “Figure 1”.

  • Figure 1: This figure is very helpful to visualize your data. I’d like to see this figure expanded or additional figures included to provide visualization of TNF-alpha concentration for all of the subgroups (example: provide a figure for the 4 CCLR subgroups and a separate table for the 4 PL subgroups). A legend could be added to indicate where significance was noted.

Our answer: we are thankful for your comment and agree with it. Figures of most important results from all CCLR and PL subgroups were added, please see the manuscript.

  • Line 251: Typo “subrgoups”

Our answer: the grammatical error was corrected and provided as: “subgroups”.

  • Line 264-276 (lameness scores), 280-293 (pain on manipulation), 319-328 (goniometry in extension), 330-332 (girthometry), 334-352 (muscle strength): Thank you, these additions here are so helpful in understanding your data. Additional figures should be considered here as well so the reader can visualize and even better understand the level of changes (or lack of changes) seen in the data in the various groups.

Our answer: we are thankful for your comment and insights, and agree with them. Figures and tables of most important results were added, please see the manuscript.

  • Line 336: Missing a space between “CCLR-P1” and “up”.

Our answer: we are grateful for your comment, the grammatical error was corrected.

  • Table 1: In the CCLR groups column, there are some typos where “CCLR-C2” was mistyped as “CCLT-C2”. Please correct.

Our answer: the errors were changed to: “CCLR-C2”.

  • Table 1 and Table 2: Thank you for changing the units of girthometry but I think you actually mean “cm” instead of “mm”. I am still struggling with the use of this outcome measure, especially given that you did not find any meaningful results, and I would strongly urge you to remove this from the manuscript. First of all, the numbers you reported in the table don’t seem reasonable given the size of dogs you reported were in the study. For example, thigh circumference of ~75 (assuming cm not mm) would be a massive dog, especially assuming we have muscle atrophy occurring given the presence of a complete cranial cruciate tear. Additionally, thigh circumference is not a standardized measurement and is widely variable depending on the size of each individual dog, so to look at this in groups of heterogenous dogs doesn’t make sense statistically. It would be a reasonable measure to look at changes for each individual dog which would require redoing the stats on this part.

Our answer: we are deeply thankful for your comments, insights and advices. We agree with them and because there is a high possibility for our results from girthometry measuerements to be misleading and due to no significant difference of them, we decided to remove it from the manuscript. All the information about girthometry was removed, please see the manuscript.

  • Line 372: Typo “There as…” should be “There is…”

Our answer: the error was corrected to: “There is a lack of research and scientific evidence of success and comparison between clinical outcomes of single intra-articular PRP injection usage and conservative treatment options of CCLR and PL in dogs.”

  • Line 380: It is misleading to discuss the “long-term activity” of PRP in this way. From your results, it is fair to say that the study supported the hypothesis of the benefits of a single PRP injection over NSAIDs over a 28 day period. Maybe use the word “sustained” instead of “long-term”.

Our answer: we are thankful for your insights and agree with them. Possible misleading statement was changed as follows: “The results of the study supports the hypothesis of the sustained benefits lasting up to 28 days after a single intra-articular PRP injection versus non-surgical management of PL and CCLR by using oral NSAIDs up to 14 days.”

  • Line 429-436: Some English editing is needed in this added paragraph more so than throughout the rest of the manuscript.

Our answer: we are thankful for your insights. The paragraph was rewritten as follows: “This study has several limitations related to the lack of data about the length of continued lameness before the initial visit, tracking and recording minor variations in treatment schemes, and tracking of animal activity during the research period. The missing data would have provided a better understanding of the acting mechanisms and outcomes of a single intra-articular PRP injection compared with the usage of oral NSAIDs in treating cases of canine CCLR and PL. Furthermore, due to our biomedical study’s nature and ethical concerns, it was impossible to use a placebo group as a control group.

            Larger sample sizes, frequent follow-up periods, and study randomization would benefit further studies. However, we believe that the results of this study may be helpful for further understanding of managing canine orthopedic diseases by using newel treatment methods.”

  • Line 436: PRP is not “curing” canine orthopedic disease. Please edit.

Our answer: it was corrected to: “However, we believe that the results of this study may be helpful for further understanding of managing canine orthopedic diseases by using newel treatment methods.”

  • Line 445. Period missing at the end of the sentence.

Our answer: we are grateful for your comment, the grammatical error was corrected.

            We hope that our changes made for the manuscript will meet your requirements and fulfills them. We are open to any further discussion or improvement of this manuscript – we believe that this research could help veterinarians to better understand usage of platelets rich products, so we are eager to improve our research as much as possible. We are immensely grateful for your help by providing insights and comments which we appreciate.